# Construction of a competency evaluation index system for front-line nurses during the outbreak of major infectious diseases: A Delphi study

**Xue Bai[1], Xiuni Gan[1]\*, Ruiqi Yang[2], Chuanlai Zhang[2], Xiaoqin Luo[3], Chengqin Luo[4], Senlin Chen[5]**

1 Department of Nursing, The Second Hospital of Chongqing Medical University, Chongqing, China,
2 Department of Intensive Care Unit, The Second Hospital of Chongqing Medical University, Chongqing, China, 3 Department of Respiratory Medicine, The Second Hospital of Chongqing Medical University, Chongqing, China, 4 Department of Emergency Medicine, The Second Hospital of Chongqing Medical University, Chongqing, China, 5 Department of Endocrinology and Breast Surgery, The First Hospital of Chongqing Medical University, Chongqing, China

\* ganxn@163.com

**Data Availability Statement:** All relevant data are within the paper and its Supporting information files.

## Abstract

### Introduction

As the frequency of infectious diseases rises, it's more important than ever to pay attention to the competency level of front-line nurses as the primary force in front-line rescue, which has an impact on the quality of anti-epidemic response. This paper aims to construct the competency evaluation index system for front-line nurses during the outbreak of major infectious diseases.

### Materials and methods

This study combined literature review, critical incident technique interviews, and semi-structured in-depth interviews, as well as two rounds of Delphi expert correspondence, to construct a competence evaluation index system for front-line nurses during the outbreak of major infectious diseases. The study used purposive sampling to select 26 experts from 11 provinces and cities across China to conduct two rounds of Delphi expert consultation, and the indicators were selected based on the mean importance score > 3.5 and the coefficient of variation < 0.25, and the weights of the indicators were calculated by the Analytic Hierarchy Process. The effective recovery rates of the two rounds of correspondence questionnaires were 93.1% and 96%.

### Results

The effective recovery rates of the two rounds of correspondence questionnaires were 93.1% and 96%, the authority coefficients of experts were 0.96 and 0.98, the Kendall's coordination coefficients of the first, second, and third level indexes were 0.281, 0.132, and 0.285 (P < 0.001), 0.259, 0.158, and 0.415 (P < 0.001). The final index system includes 4

**Funding:** This work was supported by Chongqing Science and Health Joint Medical Research Project (2020FYYX011) and The Chinese Nursing Association's 2020 Annual Project on Scientific Research (ZHKY202004). The funders had no role in study design, data collection and analysis, decision to publish, or preparation of the manuscript.

**Competing interests:** The authors have declared that no competing interests exist.

primary indicators (Knowledge System of Infectious Diseases, Nursing Skills for Infectious Diseases, Related Professional Abilities for Infectious Diseases, and Comprehensive Quality), 10 secondary indicators, and 64 tertiary indicators.

## Conclusion

The competency evaluation index system of front-line nurses during the outbreak of major infectious diseases is scientific, reasonable, and practical, which can provide a scientific basis for nursing managers to accurately understand, describe, analyze, and evaluate the competence level of nursing staff and scientifically implement the allocation of human resources in the future, as well as serve as a content framework for subsequent training programs.

## Introduction

Public health events have often occurred, including Severe Acute Respiratory Syndrome (SARS), human avian influenza, influenza A (H1N1), avian influenza A (H7N9), Ebola hemorrhagic fever, the Middle East Respiratory Syndrome (MERS), and Novel Coronavirus Disease 2019 (COVID-19) in 2019 have appeared in recent years [1]. These infectious diseases are not only dangerous to public health, but they also harm the economy and society. Nurses are the main force in the rescue of public emergencies, and their core emergency response capability has a direct impact on the quality of the overall medical rescue, which is critical for public health, social stability, and economic growth [2]. The health emergency response capability of medical personnel is primarily reflected in their competency, and those with high competency quality can provide technical-intellectual support for medical rescue with maximum efficacy, ensuring smooth rescue and reducing disability and mortality [3]. Front-line nursing staff in anti-epidemic positions should not only have solid professional knowledge and skills but also be able to provide relevant protection, emergency rescue, physical and mental stress, and care in epidemic emergencies if we want to effectively respond to infectious disease epidemics.

## Background

Although studies have been developed in recent years to evaluate the competency of health personnel at different levels during public disaster emergencies, few studies have focused in-depth on front-line nursing personnel fighting epidemics during major infectious disease epidemics. Current research on nurse competency evaluation has focused on nursing students [4], specialist nurses [5], and nursing managers [6] in routine clinical practice, and the results obtained were only on the competencies required in routine situations and did not address the competencies required during major infectious disease epidemics. Ablah et al. [7] used systematic evaluation and Delphi methods to develop a competency quality evaluation model for medical personnel for establishing a training program for competency in public health emergencies. The model includes 18 competencies in 4 major learning domains, which provides a reference value for the development of related training courses. Kan et al. [8] constructed a core emergency response capacity indicator system for healthcare workers during infectious disease emergencies in 2018, which included three primary indicators (emergency prevention capabilities for infectious disease outbreaks, infectious disease outbreak preparedness capacity, and infectious disease outbreak rescue capacity), 11 secondary indicators, and 38 tertiary

indicators. Ma et al. [9] constructed a core competency evaluation index system for advance nursing staff in emergency response to public health emergencies, including three primary indicators (first aid nursing skills, infectious disease-related knowledge, and emergency response capabilities), 12 secondary indicators, and 52 three indicators. Lin et al. [10] developed an evaluation index system for community nurses' rescue ability to respond to public health emergencies, which included four primary indicators (public health emergency prevention capacity, public health emergency rescue preparedness, rescue capacity for public health emergencies, and recovery capacity for public health emergencies), 15 secondary indicators, and 51 tertiary indicators. However, a review of the above studies reveals that the constructed competency assessment tools are not specific to the competency characteristics of nursing staff in the context of special public health events such as infectious disease epidemics, and there are fewer studies on the competency evaluation system of front-line nursing staff to combat major infectious disease epidemics. In practice, once a severe infectious disease epidemic, such as COVID-19, happens, medical institutions across the country must deploy nursing manpower in a timely and emergency way, as well as create a manpower reserve to ensure front-line human resources. However, due to the lack of a unified nursing job competency evaluation tool for major infectious disease outbreaks, the criteria for nurses to join the pool are mostly based on the principle of "department-based and consistent work," which takes into account basic factors like title, years of experience, and age [11, 12]. However, this approach makes it difficult to scientifically and systematically assess the level of nurses' competence.

With the increased frequency of public disasters in recent years, such as outbreaks of infectious disease epidemics, it is critical to construct a competency evaluation index system for front-line nurses during the outbreak of major infectious diseases that fulfills the job characteristics of front-line nursing personnel in the battle against epidemics. Thus, this study aims to:

- Construction of a competency evaluation index system for front-line nurses during the outbreak of major infectious diseases.

- Provide the scientific foundation for nursing managers to accurately understand, describe, analyze, and evaluate the competency level of front-line nurses and scientifically implement the allocation, training, and use of human resources.

- Provide a theoretical direction for the training of anti-epidemic nursing staff.

## Materials and methods

### Design

To create a draft of the index system, we used qualitative study and literature research. The Delphi study was carried out to investigate expert evaluation to competency evaluation index system for front-line nurses during the outbreak of major infectious diseases from August to October 2021.

The Delphi study is an anonymous feedback approach in which the experts do not know each other and are unable to exchange opinions during the procedure. Questionnaires with the questions to be asked to obtain expert viewpoints are used in multiple rounds of consultation until an agreement is established [13]. This method of individual consultation can prevent them from discussing or exchanging ideas [14]. As a result, the strategy is thought to be effective. We also formed a research team consisting of one professor, three nurse leaders, two postgraduate students, and one graduate students. They are in charge of reviewing literature, selecting specialists, sending out and collecting correspondence questionnaires, as well as organizing and analyzing proposals and data.

## Participants

In conducting the Delphi study, the number of experts ranged from 5 to 20, 15 to 60, or even more than 1,000 [15]. In this study, purposive sampling was used to select 29 experts, and a total of 27 experts responded to the questionnaire given to 29 experts in the first round, with a response rate of 93.1%. In the second round, questionnaires were given to 27 experts, and a total of 26 experts responded, with a response rate of 96%. The subject finally selected 26 experts from 11 provinces and cities, including Chongqing, Beijing, Liaoning, Sichuan, Shandong, Hubei, and Zhejiang and other provinces. Experts were chosen based on the following criteria: (I) bachelor's degree or above; (II) intermediate level certificate or above; (III) medical and nursing specialists who undertook COVID-2019 or other infectious diseases epidemic prevention; (IV) voluntary participation.

## Data collection

**Construction an evaluation index system.**   In this study, an interview outline was initially developed based on the STAR tool [16], literature review [17, 18], competency iceberg model [19], and research objectives, and interviews were conducted with nursing managers and nurses who had worked in anti-epidemic work after discussion in the subject group. The inclusion criteria for front-line nursing managers were: (I) managers who undertook the front-line nursing management of the COVID-19 or other infectious diseases anti-epidemic; (II) intermediate level certificate or above; (III) willing to fully express their real experiences and feelings about the front-line nursing management and the competency of front-line nursing staff. Inclusion criteria for front-line nursing staff: (I) have completed COVID-19 or other infectious diseases anti-epidemic front-line nursing work; (II) have a nurse practitioner certificate; (III) willing to fully express their true experience and feelings about front-line nursing job competency. We conducted semi-structured interviews and critical incident interviews with 13 nursing managers and 11 nurses who had participated in the anti-epidemic in eight hospitals in Chongqing from March 2021 to May 2021. The outline of the interviews is shown in S1 File. The researcher obtained informed consent from the respondents before the interviews and converted the recordings into textual information within 48 hours of the interview.

The data was organized and analyzed with Nvivo 12.0, and the competency elements were extracted using inductive content analysis [20]. Finally, a competency evaluation index system containing 4 primary indicators, 12 secondary indicators, and 67 tertiary indicators was initially constructed by combining literature research and qualitative research.

**The first draft of the expert correspondence questionnaire.**   The expert correspondence questionnaire included four parts: (I) an introduction to the study, including the research background, purpose and significance, instructions for filling out the questionnaire, and the identity of the researcher; (II) basic information of experts: gender, age, degree, title, and professional field, etc.; (III) the competency evaluation index system of front-line nurses during the outbreak of major infectious diseases expert consultation form: the importance of the index was as assessed by way of the Likert 5-level scoring method, very important = 5, relatively important = 4, general = 3, less important = 2, unimportant = 1. At the same time, experts could make suggestions for additions, deletions, and modifications; (IV) experts' familiarity with the study content and the index judgment.

**Delphi consulting and feedback cycle.**   To begin with, a correspondence questionnaire was circulated and collected by email or WeChat after the experts gave their informed consent. Meanwhile, experts scored and revised each item. After the first round of questionnaire collection, the team members summarized and analyzed the experts' opinions, and made adjustments to the indicators after discussion based on the mean value assignment > 3.5 and

coefficient of variation < 0.25 [21]. After the revision, the second round of expert consultation was conducted, and the first round of expert consultation was attached. After the second round of consultation, the results were discussed, counted, and analyzed again to finalize the index system.

## Ethical

The study was conducted in line with the guidelines of the Declaration of Helsinki and was authorized by the Ethics Committee of the Second Affiliated Hospital of Chongqing Medical University, Chongqing, China (Protocol code 114; approved on 4 November 2020). Before the survey, the experts were informed of the study's goal and verbal consent was gained from them. Participants could exit and withdraw from the survey at any time during the survey.

## Quality control

To ensure the representativeness and reliability of the study results, we used inclusion criteria to select experts. Therefore, 26 experts from 11 provinces and cities, including Chongqing, Beijing, Liaoning, Sichuan, Shandong, Hubei, and Zhejiang, etc., were selected. The research team summarized and analyzed the experts' opinions. We used Kendall's coordination coefficient and chi-square value to test the significance of experts' opinions and assure the results' reliability during the data analysis procedure.

## Data analysis

SPSS 25.0 statistical software was used to analyze the data. The measurement data were described by mean ± standard deviation, and the count data were described by frequency and percentage. The enthusiasm of the experts was expressed as the effective recovery rate of the questionnaire, the expert authority coefficient was expressed as the mean of judgment coefficient and familiarity coefficient, and the degree of expert opinion coordination was expressed as the Kendall's harmony coefficient, Kendall's coordination coefficient ranges between 0 (no agreement) and 1 (complete agreement), and the difference was considered statistically significant at $P < 0.05$. The weights of the indicators were calculated by the Analytic Hierarchy Process.

# Results

## Experts' fundamental information

According to the inclusion criteria of experts, this study finally, using a purposive sampling method, selected 26 experts from 11 provinces and cities in China, including Chongqing, Beijing, Liaoning, Sichuan, Shandong, Hubei, and Zhejiang, etc. All the experts had experience in infectious disease rescue. Their average age was 44.42 (SD 7.00) years old. Their average working years was 23.15 (SD 9.06) years. The demographic information of the experts is shown in the Table 1.

## Experts' enthusiasm

The motivation of the experts was assessed based on the return rate of the questionnaires. In the first round, 29 questionnaires were distributed and 27 effective questionnaires were returned, with a return rate of 93.1%; in the second round, 27 questionnaires were distributed and 26 effective questionnaires were returned, with a return rate of 96%.

**Table 1. Demographic information of experts.**

| Categories | Project | Frequency (N) | Proportion (%) |
|---|---|---|---|
| Gender | Male | 1 | 3.85 |
| | Female | 25 | 96.15 |
| Age (years) | 30–39 | 8 | 30.77 |
| | 40–49 | 12 | 46.15 |
| | ≥50 | 6 | 23.08 |
| Work years (years) | 3–9 | 2 | 7.69 |
| | 10–19 | 6 | 23.08 |
| | 20–29 | 10 | 38.46 |
| | ≥30 | 8 | 30.77 |
| Title | Intermediate level | 4 | 15.38 |
| | Associate senior level | 8 | 30.77 |
| | Senior level | 14 | 53.85 |
| Degree | Undergraduate | 12 | 46.15 |
| | Master | 12 | 46.15 |
| | Doctor | 2 | 7.69 |
| Research field | Critical nursing (medical) | 13 | 50.00 |
| | Infectious disease | 2 | 7.69 |
| | Public health | 1 | 3.85 |
| | Emergency nursing | 2 | 7.69 |
| | Nurse management | 8 | 30.77 |
| Position | Director of nursing department/ Department head | 6 | 23.08 |
| | Head nurse | 14 | 53.85 |
| | Others | 6 | 23.08 |
| Is it a postgraduate supervisors | Yes | 14 | 53.85 |
| | No | 12 | 46.15 |

## Expert authority coefficient and the degree of opinion coordination

In the two rounds of expert consultation, the authority coefficients were 0.96 and 0.98, respectively, which met the criteria of the expert consultation authority coefficient > 0.7 [22]. The Kendall's coordination coefficients of the first, second, and third level indicators were 0.281, 0.132, and 0.285, respectively, in the first round of expert consultation (Table 2). The Kendall's coordination coefficients of the first, second, and third level indicators were 0.259, 0.158, and 0.415, respectively, in the second round of expert consultation (Table 3). The Kendall's test had statistical significance (all p<0.001).

## The competency evaluation index system for front-line nursing staff during the outbreak of major infectious diseases

The Delphi method was used to conduct two rounds of consultation in this study. In the first round, the research team amended 13 indicators, removed 4 indicators, merged 2 indicators,

**Table 2. The result of expert suggestions' coordination degree (the first round).**

| Hierarchical indicator | Kendall's W | $X^2$ | P |
|---|---|---|---|
| primary indicators | 0.281 | 22.722 | <0.001 |
| Secondary indicators | 0.132 | 39.190 | <0.001 |
| tertiary indicators | 0.285 | 507.651 | <0.001 |

**Table 3. The result of expert suggestions' coordination degree(the second round).**

| Hierarchical indicator | Kendall's W | $X^2$ | P |
|---|---|---|---|
| primary indicators | 0.259 | 20.182 | <0.001 |
| Secondary indicators | 0.158 | 37.009 | <0.001 |
| tertiary indicators | 0.415 | 755.308 | <0.001 |

and added 6 indicators based on the exclusion criteria and experts' opinions, combining them with the discussion of the study group. In the second round, the research team changed 7 indicators, eliminated 6 indicators, and merged 1 indicator. After two rounds of expert consultation, an indicator system containing 4 primary indicators, 10 secondary indicators, and 64 tertiary indicators was finally formed. The first level indicators include those referring to the Knowledge System of Infectious Diseases, Nursing Skills for Infectious Diseases, Related Professional Abilities for Infectious Diseases, and Comprehensive Quality. The Analytic Hierarchy Process was used to calculate the index weights. In terms of weight, Nursing Skills for Infectious Diseases had the highest weight (0.345), followed by Related Professional Abilities for Infectious Diseases (0.292), Knowledge System of Infectious Diseases (0.198), and Comprehensive Quality (0.165) had the smallest weight (Table 4).

## Discussion

### The competency evaluation index system of front-line nurses during the outbreak of major infectious diseases is scientific and reliable

Based on the competency "iceberg model," this study constructs an index system through various methods such as literature research, interviews, and the Delphi method, which includes 4 primary indicators (knowledge system of infectious diseases, nursing skills for infectious diseases, related professional abilities for infectious diseases, and comprehensive quality), 10 secondary indicators, and 64 tertiary indicators. The study system scientifically and comprehensively included the competencies necessary for nursing staff to respond to major infectious disease events. The 26 correspondence experts came from 11 provinces and cities across China, and the study area covered a wide range of areas, so this study ensured the authority and representativeness of the experts. At the same time, compared with previous studies [8, 9, 23, 24], this study presents for the first time the more specialized techniques such as ventilator, extracorporeal membrane oxygenation using and monitoring skills (ECMO), continuous renal replacement therapy skills (CRRT), prone ventilation, tracheal intubation/incision, and cricothyroid puncture, which are difficult for nurses, but it was found through the preliminary study that it is still necessary for nurses to be familiar with these techniques if they want to go to the front-line rescue, and nursing managers can improve nurses' technical level in this area by holding training courses, technical competitions, scenario simulations, or examinations in order to adequately and effectively respond to an outbreak of major infectious disease. Li et al. [25] also showed that hospital managers should pay attention to the above techniques and include them in their daily training programs to improve the effectiveness of the technical reserve and ensure that nursing staff can be proficient in case of an emergency. In addition, compared with studies such as Kan et al. [8] and Ma et al. [9], this study also emphasizes for the first time the invisible characteristics that nursing staff should possess, which have been mostly ignored in previous studies, and it is worth noting that these invisible characteristics are also one of the decisive forces influencing anti-epidemic rescue; As managers, they can examine and understand the invisible characteristics of nurses by examining their

**Table 4. Competency evaluation index system for front-line nursing staff during the outbreak of major infectious diseases.**

| Index level | Significance grade ($\bar{X} \pm S$) | Variable coefficient | Weighting targets | Combination Weighting targets |
|---|---|---|---|---|
| 1. Knowledge system of infectious diseases | 4.73±0.45 | 0.10 | 0.198 | - |
| 1.1 Basic knowledge of infectious diseases | 4.69±0.47 | 0.10 | 0.790 | 0.156 |
| 1.1.1 The concept and types of infectious diseases | 4.31±0.68 | 0.16 | 0.036 | 0.006 |
| 1.1.2 Etiology and pathogenesis of infectious diseasess | 4.08±0.82 | 0.20 | 0.031 | 0.005 |
| 1.1.3 Epidemiological characteristics of infectious diseases | 4.42±0.76 | 0.17 | 0.040 | 0.006 |
| 1.1.4 Clinical manifestations of infectious diseases | 4.69±0.47 | 0.10 | 0.090 | 0.014 |
| 1.1.5 Transmission routes of infectious diseases | 4.85±0.46 | 0.09 | 0.150 | 0.023 |
| 1.1.6 Preventive measures for infectious diseases | 4.92±0.27 | 0.05 | 0.191 | 0.030 |
| 1.1.7 Diagnostic criteria for infectious diseases | 4.54±0.65 | 0.14 | 0.061 | 0.010 |
| 1.1.8 Treatment and care of infectious diseases | 4.88±0.43 | 0.09 | 0.184 | 0.029 |
| 1.1.9 Related tests for infectious diseases | 4.54±0.51 | 0.11 | 0.057 | 0.009 |
| 1.1.10 Emergency response process for infectious diseases | 4.88±0.33 | 0.07 | 0.159 | 0.025 |
| 1.2 Related knowledge of infectious diseases | 4.46±0.51 | 0.11 | 0.210 | 0.042 |
| 1.2.1 Legal and ethical knowledge | 4.88±0.33 | 0.07 | 0.250 | 0.010 |
| 1.2.2 Knowledge of caring for complex cases of infectious diseases with combined chronic diseases | 4.96±0.20 | 0.04 | 0.750 | 0.031 |
| 2. Nursing skills for infectious diseases | 5.00±0.00 | 0.00 | 0.345 | - |
| 2.1 Protection skills of infectious diseases | 5.00±0.00 | 0.00 | 0.601 | 0.207 |
| 2.1.1 Skills for putting on and taking off protective equipment | 5.00±0.00 | 0.00 | 0.438 | 0.091 |
| 2.1.2 Hand hygiene | 4.96±0.20 | 0.04 | 0.245 | 0.051 |
| 2.1.3 Disinfection and sterilization skills | 4.88±0.33 | 0.07 | 0.179 | 0.037 |
| 2.1.4 Medical waste treatment skills for infectious diseases | 4.85±0.37 | 0.08 | 0.138 | 0.029 |
| 2.2 Critical care skills | 4.88±0.33 | 0.07 | 0.294 | 0.101 |
| 2.2.1 Cardiopulmonary cerebral resuscitation skills | 4.73±0.53 | 0.11 | 0.120 | 0.012 |
| 2.2.2 ECG monitor using and monitoring skills | 4.69±0.47 | 0.10 | 0.088 | 0.009 |
| 2.2.3 Nutrition support skills | 4.50±0.65 | 0.14 | 0.045 | 0.005 |
| 2.2.4 Hemodynamic monitoring skills | 4.50±0.71 | 0.16 | 0.048 | 0.005 |
| 2.2.5 Continuous renal replacement therapy skills | 4.38±0.80 | 0.18 | 0.039 | 0.004 |
| 2.2.6 Ventilator using and monitoring skills | 4.73±0.53 | 0.11 | 0.114 | 0.012 |
| 2.2.7 Extracorporeal membrane oxygenation using and monitoring skills | 4.58±0.70 | 0.15 | 0.062 | 0.006 |
| 2.2.8 High-flow oxygen intake device using and monitoring skills | 4.65±0.49 | 0.11 | 0.068 | 0.007 |
| 2.2.9 Defibrillator using skills | 4.69±0.55 | 0.12 | 0.097 | 0.010 |
| 2.2.10 Prone position ventilation skills | 4.54±0.65 | 0.14 | 0.059 | 0.006 |
| 2.2.11 Electrocardiography machine using and monitoring skills | 4.42±0.95 | 0.21 | 0.042 | 0.004 |
| 2.2.12 Micro pump/syringe pump/infusion pump using skills | 4.00±0.69 | 0.17 | 0.028 | 0.003 |
| 2.2.13 Cricothyroid membrane puncture skills | 3.81±0.80 | 0.21 | 0.018 | 0.002 |
| 2.2.14 Tracheal intubations/tracheostomy | 4.08±0.80 | 0.20 | 0.035 | 0.004 |
| 2.2.15 Simple respirator using skills | 4.69±0.47 | 0.10 | 0.080 | 0.008 |
| 2.2.16 Artificial airway airbag pressure measurement skills | 3.81±0.63 | 0.17 | 0.017 | 0.002 |
| 2.2.17 Airway clearance skills | 4.42±0.81 | 0.18 | 0.040 | 0.004 |
| 2.3 Basic nursing skills | 4.77±0.43 | 0.09 | 0.104 | 0.036 |
| 2.3.1 Specimen collection, preservation and transportation skills | 4.96±0.20 | 0.04 | 0.508 | 0.018 |
| 2.3.2 Blood gas analysis skills | 4.85±0.37 | 0.08 | 0.203 | 0.007 |
| 2.3.3 Arteriovenous puncture skills | 4.92±0.27 | 0.05 | 0.289 | 0.010 |
| 3. Related professional abilities for infectious diseases | 4.92±0.27 | 0.05 | 0.292 | - |
| 3.1 Psychological crisis intervention abilities | 4.77±0.51 | 0.11 | 0.305 | 0.089 |
| 3.1.1 Psychological risk identification abilities | 4.96±0.20 | 0.04 | 0.511 | 0.046 |

*(Continued)*

**Table 4.** (Continued)

| Index level | Significance grade ($\bar{X} \pm S$) | Variable coefficient | Weighting targets | Combination Weighting targets |
|---|---|---|---|---|
| 3.1.2 Psychological care abilities | 4.85±0.37 | 0.08 | 0.238 | 0.021 |
| 3.1.3 Humanistic care | 4.88±0.33 | 0.07 | 0.250 | 0.022 |
| 3.2 Emergency response abilities | 4.92±0.39 | 0.08 | 0.564 | 0.165 |
| 3.2.1 Needlestick injuries emergency treatment abilities | 4.92±0.27 | 0.05 | 0.186 | 0.031 |
| 3.2.2 Blood/body fluid exposure emergency response abilities | 4.85±0.64 | 0.13 | 0.154 | 0.025 |
| 3.2.3 Suicide emergency response abilities | 4.92±0.27 | 0.05 | 0.188 | 0.031 |
| 3.2.4 Emergency response abilities of insufficient/stopped central oxygen supply | 4.88±0.33 | 0.07 | 0.159 | 0.026 |
| 3.2.5 Unexplained fainting emergency treatment abilities | 4.81±0.40 | 0.08 | 0.123 | 0.020 |
| 3.2.6 Protective equipment breakage emergency response abilities | 4.85±0.37 | 0.08 | 0.143 | 0.024 |
| 3.2.7 Material shortage emergency response abilities | 4.50±0.65 | 0.14 | 0.047 | 0.008 |
| 3.3 Complementary response abilities for infectious diseases | 4.62±0.50 | 0.11 | 0.131 | 0.038 |
| 3.3.1 Critical thinking capabilities | 4.85±0.37 | 0.08 | 0.145 | 0.006 |
| 3.3.2 Condition observation and disposal abilities | 4.88±0.33 | 0.07 | 0.180 | 0.007 |
| 3.3.3 Self-directed learning abilities | 4.69±0.55 | 0.12 | 0.096 | 0.004 |
| 3.3.4 Triage transfer abilities | 4.69±0.55 | 0.12 | 0.095 | 0.004 |
| 3.3.5 Clerical writing abilities | 4.35±0.69 | 0.16 | 0.037 | 0.001 |
| 3.3.6 Teaching abilities | 4.38±0.75 | 0.17 | 0.038 | 0.001 |
| 3.3.7 Communication and coordination abilities | 4.77±0.51 | 0.11 | 0.121 | 0.005 |
| 3.3.8 Teamwork abilities | 4.77±0.59 | 0.12 | 0.121 | 0.005 |
| 3.3.9 Organization and management abilities | 4.69±0.55 | 0.12 | 0.097 | 0.004 |
| 3.3.10 Work experience | 4.65±0.56 | 0.12 | 0.071 | 0.003 |
| 4. Comprehensive quality | 4.58±0.50 | 0.11 | 0.165 | - |
| 4.1 Ideology and morality | 4.77±0.43 | 0.09 | 0.689 | 0.114 |
| 4.1.1 Spirit of dedication | 4.96±0.20 | 0.04 | 0.482 | 0.055 |
| 4.1.2 Spirit of prudence | 4.88±0.33 | 0.07 | 0.300 | 0.034 |
| 4.1.3 Hardworking spirit | 4.81±0.40 | 0.08 | 0.219 | 0.025 |
| 4.2 Physical and mental qualities | 4.65±0.56 | 0.12 | 0.311 | 0.051 |
| 4.2.1 Physical quality | 4.96±0.20 | 0.04 | 0.346 | 0.018 |
| 4.2.2 Stress coping abilities | 4.92±0.27 | 0.05 | 0.202 | 0.010 |
| 4.2.3 Responsibility | 4.88±0.43 | 0.09 | 0.181 | 0.009 |
| 4.2.4 Self-confidence | 4.85±0.37 | 0.08 | 0.144 | 0.007 |
| 4.2.5 Optimism | 4.81±0.49 | 0.10 | 0.127 | 0.007 |

ECG: Electrocardiograph.

evaluations by leaders, colleagues, and patients in order to pick nursing personnel with high competency levels for front-line rescue.

## Analysis on the characteristics of the competency evaluation index system of front-line nurses during the outbreak of major infectious diseases

The index system constructed in this study includes 4 primary indicators, 10 secondary indicators and 64 tertiary indicators. From large to small, the weights are nursing skills for infectious diseases, related professional abilities for infectious diseases, knowledge system of infectious diseases and comprehensive quality. "Nursing Skills for Infectious Diseases" has the largest

weight, indicating that it is crucial for nurses to master relevant nursing skills in order to be competent in front-line nursing. Among the secondary indicators, "Protection skills of infectious diseases" accounts for the largest weight, which is similar to the research results of Wu [26]. Among them, the weight of "Skills for Putting On and Taking Off Protective Equipment" is the highest, because it is a prerequisite for emergency rescue to correctly wear and take off protective equipment and protect yourself in the face of infectious disease. Simultaneously, it also advises that managers should focus on protection skills in the training of nursing staff, so as to effectively cut off the transmission path of infectious diseases and minimize the occurrence of occupational exposure events [27]. The tertiary indicator "Cardiopulmonary Cerebral Resuscitation Skills" under the secondary indicator "Critical Care Skills" has the highest weight, because cardiopulmonary cerebral resuscitation can save the patient's life and reduce mortality to the greatest extent once the patient has had cardiac arrest and respiratory arrest. The weight of "Specimen Collection, Preservation, and Transportation Skills" in the secondary index "Basic Nursing Skills" is the largest, because in order to further understand the patient's condition, nurses need to accurately collect, preserve, and transport the patient's blood, sputum, pharyngeal swab, and other specimens, so as to ensure that medical staff can timely and accurately obtain the most accurate and real-time condition dynamics.

The first level indicator "Related Professional Abilities for Infectious Diseases" ranked second, and the second level indicator "Emergency Response Abilities" had the largest weight, of which "Suicide Emergency Response Abilities" had the highest weight. This may be because the isolated disease area is in a closed and high-pressure state, the patients are separated from their relatives, and with the rapid development of infectious diseases, recurring conditions, and the absence of effective drugs, so patients are prone to anxiety, helplessness, fear, and other psychological conditions that eventually lead to suicide behavior because the patients cannot bear all kinds of pressure [28]. This demonstrates that, in the event of a severe infectious disease pandemic, nurses should be able to calmly deal with patients' suicidal conduct and save their lives in time. The tertiary indicator "Psychological Risk Identification Abilities" under the secondary indicator "Psychological Crisis Intervention Abilities" has the highest weight, because the outbreak of infectious diseases often develops rapidly and the condition is complex, patients and their families do not understand the basic knowledge of the disease, which can easily lead to patients' emotional excitement, fear, and even extreme behavior [29], so timely identification of the psychological changes of patients can provide appropriate psychological intervention and social support to help patients overcome difficulties and setbacks. The highest weight was given to the tertiary indicator "Condition Observation and Disposal Abilities" under the secondary indicator "Complementary Response Abilities for Iinfectious Diseases", which may be due to the condition of patients with sudden infectious diseases being complex and changing rapidly, and there being a shortage of nursing staff. Therefore, under the condition of limited human resources, it is an essential ability for front-line nursing staff to timely detect the changing conditions of patients and take nursing measures in advance.

The primary indicator "Knowledge System of Infectious Diseases" ranked third, and the second level indicator "Basic Knowledge of Infectious Diseases" had the largest weight, among which "Preventive Measures for Infectious Diseases" had the highest weight, indicating that it is more important for nurses to correctly prevent the further occurrence and spread of infectious diseases in the epidemic situation of infectious diseases. This suggests that taking strict preventive measures can eliminate the spread of infectious diseases at the source, thus reducing the number of infections and casualties and curbing the development of infectious diseases. The highest weight is given to the tertiary indicator "Knowledge of Caring for Complex Cases of Infectious Diseases with Combined Chronic Diseases" under the secondary indicator "Related Knowledge of Infectious Diseases," because the number of patients in the isolation

ward is large and their conditions are complicated, and there are not only patients with simple infectious diseases, but also patients with other diseases combined with infectious diseases, such as hypertension, diabetes, tumors, and other diseases, nurses need to provide corresponding individualized care, so nurses should not only master the nursing knowledge of infectious diseases but also master the nursing knowledge of other diseases.

The primary indicator "Comprehensive Quality" has the smallest weight, and the secondary indicator "Ideology and Morality" has the largest weight, among which "Spirit of Dedication" has the highest weight, indicating that experts believe that the invisible characteristics of nursing staff, such as "Spirit of Dedication," should not be ignored. Personal traits are located at the bottom of the competency model and are potential and deep-level characteristics of individuals, which are not easy to observe, develop, and change, and are the determinants of personal competence [30], suggesting that managers should pay attention to the exploration of invisible traits and provide positive guidance when selecting and training anti-epidemic nursing staff. The tertiary indicator "Physical Quality" under the secondary indicator "Physical and Mental Qualities" has the highest weight, because the workload in the front-line isolation ward is heavy, and nurses need to wear airtight protective equipment, so they are prone to breathing difficulties, dizziness, sweating, rapid heart rate, physical exertion, and other symptoms during work, so having a strong body can ensure that the subsequent anti-epidemic work is carried out smoothly and effectively.

## Significance of this study

At present, competency assessment tools have been constructed for different levels of health personnel in sudden public emergency disaster events at home and abroad, but there are many classifications of public disaster events, and few studies have systematically explored the competency of front-line nursing personnel in targeted situations, and there is a lack of unified and effective assessment tools for assessing the competency of nursing staff. The competency assessment index system of front-line nursing staff during sudden major infectious disease epidemics constructed in this study is more targeted and comprehensive compared with previous studies, which can clarify their competency requirements to understand the weaknesses of nursing personnel's competency and conduct targeted training. Meanwahile, the index system can also provide a scientific basis for nursing managers to accurately understand, describe, analyze, and evaluate the competency level of front-line nursing staff and scientifically implement the allocation of human resources in the future, as well as provide a theoretical direction for the training of anti-epidemic nursing staff.

## Conclusion

Based on the competency "iceberg model," this study constructed the competency evaluation index system of front-line nurses during the outbreak of major infectious diseases through literature research, qualitative interviews, expert correspondence, and the Analytic Hierarchy Process. The index system is scientific and reasonable, and it may be used by nursing managers to evaluate the competency level of front-line nurses, clarify the weak points of nursing staff competency, and provide targeted training for the weak points so as to improve the comprehensive competency of nursing staff. It can also provide a reference basis for nursing managers to understand, evaluate, and select nursing staff for major infectious disease emergency rescue tasks and provide a content framework for subsequent training programs. However, the initially constructed index system is still limited to the theoretical framework, and the next step could be to transform the index system into a questionnaire and conduct a large-scale application to verify its scientificity and practicality and further improve the assessment tool.

## Supporting information

**S1 File. Outline of interviews.**
(DOCX)

**S2 File. Consultation questionnaire(first round).**
(DOCX)

**S3 File. Consultation questionnaire(second round).**
(DOCX)

**S4 File. Consultation questionnaire (first round) in Chinese.**
(DOCX)

**S5 File. Consultation questionnaire (second round) in Chinese.**
(DOCX)

## Acknowledgments

Thank you to all the experts who participated in this study during their busy schedules.

## Author Contributions

**Conceptualization:** Xiuni Gan.

**Data curation:** Xue Bai.

**Formal analysis:** Xue Bai, Ruiqi Yang.

**Funding acquisition:** Xiuni Gan.

**Investigation:** Xue Bai, Chuanlai Zhang, Xiaoqin Luo.

**Methodology:** Xue Bai, Xiuni Gan.

**Project administration:** Xiuni Gan.

**Resources:** Xiuni Gan, Chuanlai Zhang, Xiaoqin Luo, Chengqin Luo.

**Software:** Xue Bai, Senlin Chen.

**Supervision:** Xiuni Gan.

**Validation:** Xiuni Gan.

**Visualization:** Xue Bai.

**Writing – original draft:** Xue Bai.

**Writing – review & editing:** Xue Bai, Xiuni Gan.

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
