## [Decision Letter · Decision Letter 0]

20 Apr 2022

PONE-D-22-02197Construction of a competency evaluation index system for front-line nurses during the outbreak of major infectious diseases: A Delphi studyPLOS ONE

Dear Dr. Gan,

Thank you for submitting your manuscript to PLOS ONE. After careful consideration, we feel that it has merit but does not fully meet PLOS ONE’s publication criteria as it currently stands. Therefore, we invite you to submit a revised version of the manuscript that addresses the points raised during the review process.

Please consider all comments

We look forward to receiving your revised manuscript.

Kind regards,

Ahmed Mancy Mosa, Ph.D.

Academic Editor

PLOS ONE

Journal Requirements:

Reviewers' comments:

Reviewer's Responses to Questions

**Comments to the Author**

1. Is the manuscript technically sound, and do the data support the conclusions?

Reviewer #1: Partly

Reviewer #2: Yes

2. Has the statistical analysis been performed appropriately and rigorously? 

Reviewer #1: Yes

Reviewer #2: Yes

3. Have the authors made all data underlying the findings in their manuscript fully available?

Reviewer #1: No

Reviewer #2: No

4. Is the manuscript presented in an intelligible fashion and written in standard English?

Reviewer #1: Yes

Reviewer #2: Yes

5. Review Comments to the Author

Reviewer #1: Dear authors, I would like to thank you for your interesting paper. I read this paper with great interest that aims to construct the competency evaluation index system for front-line nurses during the outbreak of major infectious diseases. However, I have some comments aiming to enhance your paper.

Abstract:

1- more details are needed in the method section such as data collection, response rate, statical issues, qualitative and quantitative methods, and so on.

2- Add more clinical implications such as evidence-based practice, benefits ..etc.

Introduction

First of all, the paper is well written. However, a lack of references supports your competency. I notice that reference numbers 4, 5, and6 are during the COVID-19 era, what did they find?? Moreover, you can go back before COVID-19 and mention similar indicators of competency for nursing.

Methods

1- Did you purposefully select 26 experts. Or you distribute your email to several experts than 26? What about the response rate?

2- Data analysis: I noticed several methods found in the result section and they did not mention in this section. For example, Please add Kendall's coordination coefficient ranges between 0 (no agreement) and 1 (complete agreement). Add Analytic Hierarchy Process to this section either.

Results

1- How do you calculate their average age since you distributed the survey with 3 categories as mentioned in the S1 Table?

2- The first paragraph was repeated what the S1 Table. Please re-write this paragraph.

3- No need for S2 Table since you wrote everything in "Experts’ enthusiasm"

4- S3 Table, separate the first round and second round in the same Table by raw.

5- "staff during sudden major": sudden word is the first time you use it. Please remove the sudden word or replace it with the previous one to be consistent.

6- S4 Table fixed the index level at the top of the column on all pages.

Discussion

1- I noticed that the authors repeated words again and again. Instead of that discuss your findings with previous studies and highlighted what are the major findings. For example: "The competency evaluation index system of front-line nurses during the outbreak of major infectious diseases is scientific". and here "The competence evaluation index system of front-line nurses during the outbreak of major infectious diseases is reliable"

2- You stated "so patients are prone to anxiety, helplessness, fear, and other psychological conditions that eventually lead to suicide behavior" Reference, please.

3- Lack of comparison with previous studies.

4- Remove all results numbers from your discussion.

Conclusion

Add more clinical implications such as evidence-based practice and the benefits of this study.

Finally

I look forward to see your revised manuscript.

Reviewer #2: Aside the non-availability of data to support authors findings owing to privacy of consulting experts, this article has been well articulated and authors have been thorough in their approaches. The construction of this competency evaluation index system for front-line nurses in the context major infectious is well carried-out. Reviewer would recommend for publication.

6. PLOS authors have the option to publish the peer review history of their article (what does this mean?). If published, this will include your full peer review and any attached files.

Reviewer #1: No

Reviewer #2: **Yes: **Hammond Yaw Addae

---

## [Author Response · Author response to Decision Letter 0]

4 Jun 2022

A rebuttal letter

Dear reviewers:

Reviewer #1: Dear authors, I would like to thank you for your interesting paper. I read this paper with great interest that aims to construct the competency evaluation index system for front-line nurses during the outbreak of major infectious diseases. However, I have some comments aiming to enhance your paper.

Abstract:

1- more details are needed in the method section such as data collection, response rate, statical issues, qualitative and quantitative methods, and so on.

Answer: 

Dear reviewers, Thank you for your advice. This study combined literature review, critical incident technique interviews, and semi-structured in-depth interviews, as well as two rounds of Delphi expert correspondence, to construct a competence evaluation index system for front-line nurses during the outbreak of major infectious diseases. The study used purposive sampling to select 26 experts from 11 provinces and cities across China to conduct two rounds of Delphi expert consultation, and the indicators were selected based on the mean importance score > 3.5 and the coefficient of variation < 0.25, and the weights of the indicators were calculated by the Analytic Hierarchy Process. The effective recovery rates of the two rounds of correspondence questionnaires were 93.1% and 96%.

2-Add more clinical implications such as evidence-based practice, benefits ..etc.

Answer: 

Dear reviewers, Thank you for your advice. The competency evaluation index system of front-line nurses during the outbreak of major infectious diseases is scientific, reasonable, and practical, which can provide a scientific basis for nursing managers to accurately understand, describe, analyze, and evaluate the competence level of nursing staff and scientifically implement the allocation of human resources in the future, as well as serve as a content framework for subsequent training programs.

Introduction

First of all, the paper is well written. However, a lack of references supports your competency. I notice that reference numbers 4, 5, and6 are during the COVID-19 era, what did they find?? Moreover, you can go back before COVID-19 and mention similar indicators of competency for nursing.

Answer: 

Dear reviewers, Thank you for your advice. Although studies have been developed in recent years to evaluate the competency of health personnel at different levels during public disaster emergencies, few studies have focused in-depth on front-line nursing personnel fighting epidemics during major infectious disease epidemics. Current research on nurse competency evaluation has focused on nursing students [4], specialist nurses [5], and nursing managers [6] in routine clinical practice, and the results obtained were only on the competencies required in routine situations and did not address the competencies required during major infectious disease epidemics. Ablah et al.[7] used systematic evaluation and Delphi methods to develop a competency quality evaluation model for medical personnel for establishing a training program for competency in public health emergencies. The model includes 18 competencies in 4 major learning domains, which provides a reference value for the development of related training courses. Kan et al.[8] constructed a core emergency response capacity indicator system for healthcare workers during infectious disease emergencies in 2018, which included three primary indicators (emergency prevention capabilities for infectious disease outbreaks, infectious disease outbreak preparedness capacity, and infectious disease outbreak rescue capacity), 11 secondary indicators, and 38 tertiary indicators. Ma et al.[9] constructed a core competency evaluation index system for advance nursing staff in emergency response to public health emergencies, including three primary indicators (first aid nursing skills, infectious disease-related knowledge, and emergency response capabilities), 12 secondary indicators, and 52 three indicators. Lin et al.[10] developed an evaluation index system for community nurses' rescue ability to respond to public health emergencies, which included four primary indicators (public health emergency prevention capacity, public health emergency rescue preparedness, rescue capacity for public health emergencies, and recovery capacity for public health emergencies), 15 secondary indicators, and 51 tertiary indicators. However, a review of the above studies reveals that the constructed competency assessment tools are not specific to the competency characteristics of nursing staff in the context of special public health events such as infectious disease epidemics, and there are fewer studies on the competency evaluation system of front-line nursing staff to combat major infectious disease epidemics.

Methods

1- Did you purposefully select 26 experts. Or you distribute your email to several experts than 26? What about the response rate?

Answer:

Dear reviewers, Thank you for your advice. In this study, purposive sampling was used to select 29 experts, and a total of 27 experts responded to the questionnaire given to 29 experts in the first round, with a response rate of 93.1%. In the second round, questionnaires were given to 27 experts, and a total of 26 experts responded, with a response rate of 96%. 

2- Data analysis: I noticed several methods found in the result section and they did not mention in this section. For example, Please add Kendall's coordination coefficient ranges between 0 (no agreement) and 1 (complete agreement). Add Analytic Hierarchy Process to this section either.

Answer:

Dear reviewers, Thank you for your advice. The degree of expert opinion coordination was expressed as the Kendall’s harmony coefficient, Kendall's coordination coefficient ranges between 0 (no agreement) and 1 (complete agreement), and the difference was considered statistically significant at P<0.05. The weights of the indicators were calculated by the Analytic Hierarchy Process.

Results

1- How do you calculate their average age since you distributed the survey with 3 categories as mentioned in the S1 Table?

Answer:

Dear reviewers, Thank you for your advice. We asked the experts to fill in their age in the first round of the correspondence questionnaire. And we are putting the real age of each expert into SPSS and then using descriptive statistical analysis to figure out the average age.

2- The first paragraph was repeated what the S1 Table. Please re-write this paragraph.

Answer:

Dear reviewers, Thank you for your advice. According to the inclusion criteria of experts, this study finally, using a purposive sampling method, selected 26 experts from 11 provinces and cities in China, including Chongqing, Beijing, Liaoning, Sichuan, Shandong, Hubei, and Zhejiang, etc. All the experts had experience in infectious disease rescue. Their average age was 44.42 (SD 7.00) years old. Their average working years was 23.15 (SD 9.06) years. The demographic information of the experts is shown in the Table 1. (note:S1 Table has been revised to Table 1)

3-No need for S2 Table since you wrote everything in "Experts’ enthusiasm."

Answer:

Dear reviewers, Thank you for your advice. Table 2 has been deleted. (note:S2 Table has been revised to Table 2)

4- S3 Table, separate the first round and second round in the same Table by raw.

Answer: 

Dear reviewers, Thank you for your advice. S3 Table has been revised to Table 2 and Table 3.

 Table 2 The result of expert suggestions’coordination degree (the first round)

Hierarchical indicator Kendall’s W X2 P

primary indicators 0.281 22.722 ＜0.001

Secondary indicators 0.132 39.190 ＜0.001

tertiary indicators 0.285 507.651 ＜0.001

 Table 3 The result of expert suggestions’coordination degree(the second round)

Hierarchical indicator Kendall’s W X2 P

primary indicators 0.259 20.182 ＜0.001

Secondary indicators 0.158 37.009 ＜0.001

tertiary indicators 0.415 755.308 ＜0.001

5- "staff during sudden major": sudden word is the first time you use it. Please remove the sudden word or replace it with the previous one to be consistent.

Answer:

Dear reviewers, Thank you for your advice. The sudden word has been deleted.

6-S4 Table fixed the index level at the top of the column on all pages.

Answer:

Dear reviewers, Thank you for your advice. Table 4 has been revised. (note:S4 Table has been revised to Table 4)

 Table 4 Competency evaluation index system for front-line nursing staff during the outbreak of major infectious diseases

Index level Significance

grade（X±S） Variable

coefficient Weighting

targets Combination Weighting targets

1.Knowledge system of infectious diseases 4.73±0.45 0.10 0.198 -

1.1 Basic knowledge of infectious diseases 4.69±0.47 0.10 0.790 0.156

1.1.1 The concept and types of infectious diseases 4.31±0.68 0.16 0.036 0.006

1.1.2 Etiology and pathogenesis of infectious diseasess 4.08±0.82 0.20 0.031 0.005

1.1.3 Epidemiological characteristics of infectious diseases 4.42±0.76 0.17 0.040 0.006

1.1.4 Clinical manifestations of infectious diseases 4.69±0.47 0.10 0.090 0.014

1.1.5 Transmission routes of infectious diseases 4.85±0.46 0.09 0.150 0.023

1.1.6 Preventive measures for infectious diseases 4.92±0.27 0.05 0.191 0.030

1.1.7 Diagnostic criteria for infectious diseases 4.54±0.65 0.14 0.061 0.010

1.1.8 Treatment and care of infectious diseases 4.88±0.43 0.09 0.184 0.029

1.1.9 Related tests for infectious diseases 4.54±0.51 0.11 0.057 0.009

1.1.10 Emergency response process for infectious diseases 4.88±0.33 0.07 0.159 0.025

1.2 Related knowledge of infectious diseases 4.46±0.51 0.11 0.210 0.042

1.2.1 Legal and ethical knowledge 4.88±0.33 0.07 0.250 0.010

1.2.2 Knowledge of caring for complex cases of infectious diseases with combined chronic diseases 4.96±0.20 0.04 0.750 0.031

2.Nursing skills for infectious diseases 5.00±0.00 0.00 0.345 -

Table 4. (Continued)

Index level Significance

grade（X±S） Variable

coefficient Weighting

targets Combination Weighting targets

2.1 Protection skills of infectious diseases 5.00±0.00 0.00 0.601 0.207

2.1.1 Skills for putting on and taking off protective equipment 5.00±0.00 0.00 0.438 0.091

2.1.2 Hand hygiene 4.96±0.20 0.04 0.245 0.051 

2.1.3 Disinfection and sterilization skills 4.88±0.33 0.07 0.179 0.037

2.1.4 Medical waste treatment skills for infectious diseases 4.85±0.37 0.08 0.138 0.029

2.2 Critical care skills 4.88±0.33 0.07 0.294 0.101

2.2.1 Cardiopulmonary cerebral resuscitation skills 4.73±0.53 0.11 0.120 0.012

2.2.2 ECG monitor using and monitoring skills 4.69±0.47 0.10 0.088 0.009

2.2.3 Nutrition support skills 4.50±0.65 0.14 0.045 0.005

2.2.4 Hemodynamic monitoring skills 4.50±0.71 0.16 0.048 0.005

2.2.5 Continuous renal replacement therapy skills 4.38±0.80 0.18 0.039 0.004

2.2.6 Ventilator using and monitoring skills 4.73±0.53 0.11 0.114 0.012

2.2.7 Extracorporeal membrane oxygenation using and monitoring skills 4.58±0.70 0.15 0.062 0.006

2.2.8 High-flow oxygen intake device using and monitoring skills 4.65±0.49 0.11 0.068 0.007

2.2.9 Defibrillator using skills 4.69±0.55 0.12 0.097 0.010

Table 4. (Continued)

Index level Significance

grade（X±S） Variable

coefficient Weighting

targets Combination Weighting targets

2.2.10 Prone position ventilation skills 4.54±0.65 0.14 0.059 0.006

2.2.11 Electrocardiography machine using and monitoring skills 4.42±0.95 0.21 0.042 0.004

2.2.12 Micro pump/syringe pump/infusion pump using skills 4.00±0.69 0.17 0.028 0.003

2.2.13 Cricothyroid membrane puncture skills 3.81±0.80 0.21 0.018 0.002

2.2.14 Tracheal intubations/tracheostomy 4.08±0.80 0.20 0.035 0.004

2.2.15 Simple respirator using skills 4.69±0.47 0.10 0.080 0.008

2.2.16 Artificial airway airbag pressure measurement skills 3.81±0.63 0.17 0.017 0.002

2.2.17 Airway clearance skills 4.42±0.81 0.18 0.040 0.004

2.3 Basic nursing skills 4.77±0.43 0.09 0.104 0.036

2.3.1 Specimen collection, preservation and transportation skills 4.96±0.20 0.04 0.508 0.018

2.3.2Blood gas analysis skills 4.85±0.37 0.08 0.203 0.007

2.3.3 Arteriovenous puncture skills 4.92±0.27 0.05 0.289 0.010

3.Related professional abilities for infectious diseases 4.92±0.27 0.05 0.292 -

3.1 Psychological crisis intervention abilities 4.77±0.51 0.11 0.305 0.089

3.1.1 Psychological risk identification abilities 4.96±0.20 0.04 0.511 0.046 

Table 4. (Continued)

Index level Significance

grade（X±S） Variable

coefficient Weighting

targets Combination Weighting targets

3.1.2 Psychological care abilities 4.85±0.37 0.08 0.238 0.021

3.1.3 Humanistic care 4.88±0.33 0.07 0.250 0.022

3.2 Emergency response abilities 4.92±0.39 0.08 0.564 0.165

3.2.1 Needlestick injuries emergency treatment abilities 4.92±0.27 0.05 0.186 0.031

3.2.2 Blood/body fluid exposure emergency response abilities 4.85±0.64 0.13 0.154 0.025

3.2.3 Suicide emergency response abilities 4.92±0.27 0.05 0.188 0.031

3.2.4 Emergency response abilities of insufficient/stopped central oxygen supply 4.88±0.33 0.07 0.159 0.026

3.2.5 Unexplained fainting emergency treatment abilities 4.81±0.40 0.08 0.123 0.020

3.2.6 Protective equipment breakage emergency response abilities 4.85±0.37 0.08 0.143 0.024

3.2.7 Material shortage emergency response abilities 4.50±0.65 0.14 0.047 0.008

3.3 Complementary response abilities for infectious diseases 4.62±0.50 0.11 0.131 0.038

3.3.1 Critical thinking capabilities 4.85±0.37 0.08 0.145 0.006

3.3.2 Condition observation and disposal abilities 4.88±0.33 0.07 0.180 0.007

3.3.3 Self-directed learning abilities 4.69±0.55 0.12 0.096 0.004

3.3.4 Triage transfer abilities 4.69±0.55 0.12 0.095 0.004

Table 4. (Continued)

Index level Significance

grade（X±S） Variable

coefficient Weighting

targets Combination Weighting targets

3.3.5 Clerical writing abilities 4.35±0.69 0.16 0.037 0.001

3.3.6 Teaching abilities 4.38±0.75 0.17 0.038 0.001

3.3.7 Communication and coordination abilities 4.77±0.51 0.11 0.121 0.005

3.3.8 Teamwork abilities 4.77±0.59 0.12 0.121 0.005

3.3.9 Organization and management abilities 4.69±0.55 0.12 0.097 0.004

3.3.10 Work experience 4.65±0.56 0.12 0.071 0.003

4.Comprehensive quality 4.58±0.50 0.11 0.165 -

4.1 Ideology and morality 4.77±0.43 0.09 0.689 0.114

4.1.1 Spirit of dedication 4.96±0.20 0.04 0.482 0.055

4.1.2 Spirit of prudence 4.88±0.33 0.07 0.300 0.034

4.1.3 Hardworking spirit 4.81±0.40 0.08 0.219 0.025

4.2 Physical and mental qualities 4.65±0.56 0.12 0.311 0.051

4.2.1 Physical quality 4.96±0.20 0.04 0.346 0.018

4.2.2 Stress coping abilities 4.92±0.27 0.05 0.202 0.010

4.2.3 Responsibility 4.88±0.43 0.09 0.181 0.009

4.2.4 Self-confidence 4.85±0.37 0.08 0.144 0.007

4.2.5 Optimism 4.81±0.49 0.10 0.127 0.007

Discussion

1- I noticed that the authors repeated words again and again. Instead of that discuss your findings with previous studies and highlighted what are the major findings. For example: "The competency evaluation index system of front-line nurses during the outbreak of major infectious diseases is scientific". and here "The competence evaluation index system of front-line nurses during the outbreak of major infectious diseases is reliable"

Answer:

Dear reviewers, Thank you for your advice. 

The competency evaluation index system of front-line nurses during the outbreak of major infectious diseases is scientific and reliable

Based on the competency "iceberg model," this study constructs an index system through various methods such as literature research, interviews, and the Delphi method, which includes 4 primary indicators (knowledge system of infectious diseases, nursing skills for infectious diseases, related professional abilities for infectious diseases, and comprehensive quality), 10 secondary indicators, and 64 tertiary indicators. The study system scientifically and comprehensively included the competencies necessary for nursing staff to respond to major infectious disease events. The 26 correspondence experts came from 11 provinces and cities across China, and the study area covered a wide range of areas, so this study ensured the authority and representativeness of the experts. At the same time, compared with previous studies[8, 9, 23, 24], this study presents for the first time the more specialized techniques such as ventilator, ECMO, CRRT, prone ventilation, tracheal intubation/incision, and cricothyroid puncture, which are difficult for nurses, but it was found through the preliminary study that it is still necessary for nurses to be familiar with these techniques if they want to go to the front-line rescue, and nursing managers can improve nurses' technical level in this area by holding training courses, technical competitions, scenario simulations, or examinations in order to adequately and effectively respond to an outbreak of major infectious disease. Li et al. [25] also showed that hospital managers should pay attention to the above techniques and include them in their daily training programs to improve the effectiveness of the technical reserve and ensure that nursing staff can be proficient in case of an emergency. In addition, compared with studies such as Kan et al. [8] and Ma et al. [9], this study also emphasizes for the first time the invisible characteristics that nursing staff should possess, which have been mostly ignored in previous studies, and it is worth noting that these invisible characteristics are also one of the decisive forces influencing anti-epidemic rescue; As managers, they can examine and understand the invisible characteristics of nurses by examining their evaluations by leaders, colleagues, and patients in order to pick nursing personnel with high competency levels for front-line rescue.

2-You stated "so patients are prone to anxiety, helplessness, fear, and other psychological conditions that eventually lead to suicide behavior" Reference, please.

Answer:

Dear reviewers, Thank you for your advice. 

Reference: 28. Sher L. The impact of the COVID-19 pandemic on suicide rates. QJM. 2020;113(10):707-712. doi: 10.1093/qjmed/hcaa202. 

3- Lack of comparison with previous studies.

Answer:

Dear reviewers, Thank you for your advice. Literature has been added, and see first paragraph of discussion.

The competency evaluation index system of front-line nurses during the outbreak of major infectious diseases is scientific and reliable

Based on the competency "iceberg model," this study constructs an index system through various methods such as literature research, interviews, and the Delphi method, which includes 4 primary indicators (knowledge system of infectious diseases, nursing skills for infectious diseases, related professional abilities for infectious diseases, and comprehensive quality), 10 secondary indicators, and 64 tertiary indicators. The study system scientifically and comprehensively included the competencies necessary for nursing staff to respond to major infectious disease events. The 26 correspondence experts came from 11 provinces and cities across China, and the study area covered a wide range of areas, so this study ensured the authority and representativeness of the experts. At the same time, compared with previous studies[8, 9, 23, 24], this study presents for the first time the more specialized techniques such as ventilator, ECMO, CRRT, prone ventilation, tracheal intubation/incision, and cricothyroid puncture, which are difficult for nurses, but it was found through the preliminary study that it is still necessary for nurses to be familiar with these techniques if they want to go to the front-line rescue, and nursing managers can improve nurses' technical level in this area by holding training courses, technical competitions, scenario simulations, or examinations in order to adequately and effectively respond to an outbreak of major infectious disease. Li et al. [25] also showed that hospital managers should pay attention to the above techniques and include them in their daily training programs to improve the effectiveness of the technical reserve and ensure that nursing staff can be proficient in case of an emergency. In addition, compared with studies such as Kan et al. [8] and Ma et al. [9], this study also emphasizes for the first time the invisible characteristics that nursing staff should possess, which have been mostly ignored in previous studies, and it is worth noting that these invisible characteristics are also one of the decisive forces influencing anti-epidemic rescue; As managers, they can examine and understand the invisible characteristics of nurses by examining their evaluations by leaders, colleagues, and patients in order to pick nursing personnel with high competency levels for front-line rescue.

4- Remove all results numbers from your discussion.

Answer:

Dear reviewers, Thank you for your advice. All numbers have been removed.

Conclusion

Add more clinical implications such as evidence-based practice and the benefits of this study.

Dear reviewers, Thank you for your advice. Based on the competency "iceberg model," this study constructed the competency evaluation index system of front-line nurses during the outbreak of major infectious diseases through literature research, qualitative interviews, expert correspondence, and the Analytic Hierarchy Process. The index system is scientific and reasonable, and it may be used by nursing managers to evaluate the competency level of front-line nurses, clarify the weak points of nursing staff competency, and provide targeted training for the weak points so as to improve the comprehensive competency of nursing staff. It can also provide a reference basis for nursing managers to understand, evaluate, and select nursing staff for major infectious disease emergency rescue tasks and provide a content framework for subsequent training programs. However, the initially constructed index system is still limited to the theoretical framework, and the next step could be to transform the index system into a questionnaire and conduct a large-scale application to verify its scientificity and practicality and further improve the assessment tool.

---

## [Decision Letter · Decision Letter 1]

20 Jun 2022

Construction of a competency evaluation index system for front-line nurses during the outbreak of major infectious diseases: A Delphi study

PONE-D-22-02197R1

Dear Dr. Gan,

We’re pleased to inform you that your manuscript has been judged scientifically suitable for publication and will be formally accepted for publication once it meets all outstanding technical requirements.

Kind regards,

Ahmed Mancy Mosa, Ph.D.

Academic Editor

PLOS ONE

Additional Editor Comments (optional):

Reviewers' comments:

Reviewer's Responses to Questions

**Comments to the Author**

1. If the authors have adequately addressed your comments raised in a previous round of review and you feel that this manuscript is now acceptable for publication, you may indicate that here to bypass the “Comments to the Author” section, enter your conflict of interest statement in the “Confidential to Editor” section, and submit your "Accept" recommendation.

Reviewer #1: All comments have been addressed

2. Is the manuscript technically sound, and do the data support the conclusions?

Reviewer #1: Yes

3. Has the statistical analysis been performed appropriately and rigorously? 

Reviewer #1: Yes

4. Have the authors made all data underlying the findings in their manuscript fully available?

Reviewer #1: Yes

5. Is the manuscript presented in an intelligible fashion and written in standard English?

Reviewer #1: Yes

6. Review Comments to the Author

Reviewer #1: Dear author,

You have edited your article and answered my questions and your manuscript has improved greatly. I have minor comments that would also strengthen you article.

1- All abbreviation in the Tables should be explained in footnote.

2- All abbreviation in text should be stated clearly such as ECMO, CRRT. Its a matter for non medical readers.

3 Make sure that your line paragraph spacing is 2. Please see the paragraph under heading participants.

Good luck and congratulations

7. PLOS authors have the option to publish the peer review history of their article (what does this mean?). If published, this will include your full peer review and any attached files.

Reviewer #1: **Yes: **Othman A. Alfuqaha

---

## [Editor Report · Acceptance letter]

23 Jun 2022

PONE-D-22-02197R1 

Construction of a competency evaluation index system for front-line nurses during the outbreak of major infectious diseases: A Delphi study 

Dear Dr. Gan:

I'm pleased to inform you that your manuscript has been deemed suitable for publication in PLOS ONE. Congratulations! Your manuscript is now with our production department. 

Kind regards, 

on behalf of

Dr. Ahmed Mancy Mosa 

Academic Editor

PLOS ONE